# Sarcopenia as a Predictor of Short- and Long-Term Outcomes in Patients Surgically Treated for Malignant Pleural Mesothelioma

**DOI:** 10.3390/cancers14153699

**Published:** 2022-07-29

**Authors:** Eleonora Faccioli, Stefano Terzi, Chiara Giraudo, Andrea Zuin, Antonella Modugno, Francesco Labella, Giovanni Zambello, Giulia Lorenzoni, Marco Schiavon, Dario Gregori, Giulia Pasello, Fiorella Calabrese, Andrea Dell’Amore, Federico Rea

**Affiliations:** 1Thoracic Surgery Unit, Department of Cardiac, Thoracic Vascular Sciences and Public Health, University Hospital of Padova, 35128 Padova, Italy; ste.terzi@gmail.com (S.T.); andrea.zuin@unipd.it (A.Z.); giovanni.zambello@aopd.veneto.it (G.Z.); marco.schiavon@unipd.it (M.S.); andrea.dellamore@unipd.it (A.D.); federico.rea@unipd.it (F.R.); 2Radiology Unit, Department of Medicine, University Hospital of Padova, 35128 Padova, Italy; chiara.giraudo@unipd.it (C.G.); antonella.modugno@aopd.veneto.it (A.M.); francesco.labella@aopd.veneto.it (F.L.); 3Biostatistics Unit, Department of Cardiac, Thoracic Vascular Sciences and Public Health, University Hospital of Padova, 35128 Padova, Italy; giulia.lorenzoni@unipd.it (G.L.); dario.gregori@unipd.it (D.G.); 4Medical Oncology 2, Istituto Oncologico Veneto IOV-IRCCS, 35128 Padova, Italy; giulia.pasello@iov.veneto.it; 5Pathology Unit, Department of Cardiac, Thoracic Vascular Sciences and Public Health, University Hospital of Padova, 35128 Padova, Italy; fiorella.calabrese@unipd.it

**Keywords:** mesothelioma, sarcopenia, surgery, survival

## Abstract

**Simple Summary:**

Malignant pleural mesothelioma (MPM) is an aggressive asbestos-related tumor with a poor prognosis. Surgery, often considered in the context of multimodality treatment, may be burdened by high morbidity, and for this reason, it should be reserved for patients who have a good pre-operative performance status. Sarcopenia, a well-established predictor of negative outcomes in several clinical settings, is still underinvestigated in MPM. The aim of the study is to elucidate the prognostic impact of muscular loss on surgical outcomes in patients with MPM. We demonstrated that, respectively, pre- and post-operative sarcopenia strongly affects the risk of post-operative complications and long-term survival after surgery for MPM. This finding will help clinicians to perform a better selection of patients, taking into consideration the enrollment in dedicated rehabilitation programs before surgery.

**Abstract:**

Surgery for malignant pleural mesothelioma (MPM) should be reserved only for patients who have a good performance status. Sarcopenia, a well-known predictor of poor outcomes after surgery, is still underinvestigated in MPM. The aim of this study is to evaluate the role of sarcopenia as a predictor of short-and long-term outcomes in patients surgically treated for MPM. In our analysis, we included patients treated with a cytoreductive intent in a multimodality setting, with both pre- and post-operative CT scans without contrast available. We excluded those in whom a complete macroscopic resection was not achieved. Overall, 86 patients were enrolled. Sarcopenia was assessed by measuring the mean muscular density of the bilateral paravertebral muscles (T12 level) on pre-and post-operative CTs; a threshold value of 30 Hounsfield Units (HU) was identified. Sarcopenia was found pre-operatively in 57 (66%) patients and post-operatively in 61 (74%). Post-operative sarcopenic patients had a lower 3-year overall survival (OS) than those who were non-sarcopenic (34.9% vs. 57.6% *p* = 0.03). Pre-operative sarcopenia was significantly associated with a higher frequency of post-operative complications (65% vs. 41%, *p* = 0.04). The evaluation of sarcopenia, through a non-invasive method, would help to better select patients submitted to surgery for MPM in a multimodality setting.

## 1. Introduction

Malignant Pleural Mesothelioma (MPM) is a relatively rare tumor characterized by a locally aggressive behavior that leads to a fatal prognosis mostly due to (i) relative chemo- and radio-resistance, (ii) difficulties in obtaining a radical excision with surgery, and (iii) the frequency advanced disease at the time of diagnosis. Despite the fact that surgery plays an important role in MPM management, the best combination of the available therapeutical options (surgery with or without chemotherapy and/or radiotherapy both in adjuvant and/or neoadjuvant settings) is still controversial. Even with the multidisciplinary approach, the prognosis is poor, with a reported median overall survival (OS) between 9 and 17 months [1]. For this reason, it is important to adequately select patients who might be the most appropriate candidates for surgery.

Sarcopenia is defined as a syndrome characterized by a progressive and generalized loss of skeletal muscle mass and function (strength or performance) with a risk of adverse outcomes such as physical disability, poor quality of life, and death [2]. However, the loss of muscle mass is actually a matter of debate concerning definition, cut-off points, and methods of measurement. Currently, a wide variety of tests and tools are available for the definition and characterization of sarcopenia both in clinical and research fields. Sarcopenia can be evaluated by measuring muscle strength, for example, with a hand grip test and a chair stand test, by measuring muscle quantity using computed tomography and magnetic resonance imaging, and by measuring physical performance with, for example, the gait speed and walking test [2].

Significant muscle loss is one of the most prevalent and serious cancer-related events strongly correlated to a poor prognosis [3], especially in patients with advanced disease. Several studies have recently reported that sarcopenia in patients with gastric [4], colorectal [5], pancreatic [6], hepatocellular [7], endometrial [8], and renal cell [9] carcinomas is associated with poor post-operative survival. In thoracic surgery, low muscle mass has been investigated in lung cancer [10] and also in lung transplantation [11,12,13], and, in both of these clinical settings, it is related to a poor prognosis.

However, in the current literature, there is a lack of evidence about the association between sarcopenia and surgically-treated MPM and its impact on clinical outcomes.

Thus, in the present single-center retrospective study, we aimed to evaluate if sarcopenia, assessed with Computed Tomography (CT), could be a predictive factor of long-and short-term outcomes in patients affected by MPM treated by surgery.

## 2. Materials and Methods

### 2.1. Study Population

Between July 1994 and July 2021, 305 patients were surgically treated for MPM at the Thoracic Surgery Unit of the University Hospital of Padua, Italy. Among this population, we retrospectively collected data from patients who underwent surgery with curative intent by pleurectomy decortication (PD, parietal and visceral pleurectomy without the resection of the diaphragm or pericardium), extended pleurectomy-decortication (EPD, parietal and visceral pleurectomy with the removal of the diaphragm and/or pericardium), or extra-pleural pneumonectomy (EPP, en bloc resection of the visceral and parietal pleura, lung and, if necessary, ipsilateral diaphragm and pericardium). The surgical procedure was always performed in a setting of multimodal treatment, which consisted of neoadjuvant chemotherapy (CT) and adjuvant radiotherapy (RT). After surgery, according to our protocol, at our Institution, all of the patients are enrolled in a dedicated rehabilitation protocol: the day after surgery, they are evaluated by a physiatrist who sets a specific respiratory program tailored to every single patient. For the subsequent days, the patients are followed by physiotherapists to improve physical and respiratory conditions, especially with the use of an incentive spirometer, and instructed to continue these exercises after discharge.

We otherwise excluded those patients who were operated on for palliative intent or those in whom a complete macroscopic resection could not be achieved. Again, to avoid bias as much as possible, for the analysis, we decided to enroll only patients with both pre- and post-operative CT scans without contrast and with a homogeneous slice thickness (3 mm) available in our radiological archives. Finally, 86 patients satisfied the criteria to be included in the study (Figure 1). For every patient demographic, clinical and radiological data, as well as information on multimodality treatments and outcomes, were collected. All of the data were abstracted from electronic medical records and elaborated in an excel database. The study protocol was approved by the Ethics Committee of the University Hospital of Padua (n. PD732-2220T). Due to the retrospective nature of the study, written informed consent was waived. The study was conducted in accordance with The Code of Ethics of the World Medical Association (Declaration of Helsinki).

### 2.2. Radiological Evaluation of the Sarcopenia

Pre- and post-operative CT scans without contrast and with a slice thickness of 3 mm were analyzed by two dedicated pulmonary radiologists with more than ten years of experience in musculoskeletal imaging. For each patient, the mean Hounsfield Unit (HU) value of the bilateral paravertebral muscles at the level of the 12th thoracic vertebra, using a free-hand region of interest, was calculated. All of the measures were performed with open-source software (Horos v3, www.horosproject.org (accessed between 1 January 2021 and 31 March 2021). According to the literature, a value of 30 HU was assessed as a cut-off to define a patient as sarcopenic [14,15,16,17]. Figure 2 shows how the radiological evaluation in a non-sarcopenic (a) and in a sarcopenic patient, respectively, (b) was performed.

### 2.3. Statistical Analysis

Descriptive statistics were reported as median with I and III interquartile range for continuous variables and as absolute numbers and relative frequencies in the case of categorical variables.

To analyze the post-operative outcomes, Gamma models and logistic regression for continuous and categorical variables, respectively, were employed.

The Kaplan–Meier approach was used to evaluate the survival distribution, while the disease recurrence was evaluated using the Cumulative Incidence Function (CIF) to account for competing risks.

Cox regression models were estimated to assess the effect of covariates on survival and relapse. The results were reported as Hazard Ratio (HR), 95% Confidence Interval (CI), and *p*-value. Restrictive Cubic Spline (RCS) was employed to model non-linear associations.

The analysis was performed using R software (R Core Team, http://www.r-project.org/index.html (accessed between 1 January 2021 and 31 March 2021) within rms, survival, and cmprsk packages.

## 3. Results

### 3.1. Demographic and Clinical Data

Overall, 86 patients (76% males) with a median age of 66 years were enrolled in the study. The majority of them (65%) were submitted to a right-sided surgical procedure which in 62 cases (72%) was a PD or an EPD and in less than 30% an EPP. The entire population (100%) underwent neo-adjuvant chemotherapy, while 72 patients (84%) were submitted to adjuvant radiotherapy as a part of the standard multimodality treatment practiced for MPM at our Institution. Sarcopenia, assessed with paravertebral muscle densitometry at T12 level and using 30 HU as cut-off, was found pre-operatively in 57 patients (66%) with a median value of 27 HU and post-operatively in 63 (74%) with a median value of 25 HU. Median overall survival (OS) and disease-free survival (DFS) were respectively 21 and 13 months. All of the demographic and clinical data of the overall population are reported in Table 1.

In Table 2 and Table 3, respectively, the data on pre- and post-operative sarcopenic patients are reported and compared to non-sarcopenic patients. No significant differences were found between the groups except for older age in pre- and post-operative sarcopenic patients (*p* = 0.022, *p* = 0.014), a higher frequency of asbestos exposure in patients with pre-operative sarcopenia (*p* = 0.037), and, as expected, a lower density of paravertebral muscles in sarcopenic patients (*p* < 0.001).

### 3.2. Post-Operative Outcomes

In Table 4, the data on post-operative outcomes concerning the length of hospital stay and complications are reported for patients with pre-operative sarcopenia compared to those without this pre-operative condition. The patients with pre-operative sarcopenia were more likely to experience post-operative complications (*p* = 0.04) without affecting the length of hospital stay, as the majority of them (73%) required only a pharmacological treatment to be solved.

### 3.3. Analysis of Overall Survival and Cumulative Incidence of Relapse

Comparing the overall survival (OS) rates, we found that OS was significantly lower in patients affected by post-operative sarcopenia than in those who were not sarcopenic (HR 1.96, 1.06–3.64, *p* = 0.03). The 3-year survival rate was, respectively, 34.9% vs. 57.6% (Figure 3).

On the other hand, the 3-year cumulative incidence of relapse between patients affected by post-operative sarcopenia and those with preserved muscular mass was, respectively, 81% vs. 74% (*p* = 0.70), showing no significant differences (Figure 4).

Furthermore, in the univariable analysis, mortality at follow-up was significantly associated with post-operative sarcopenia (HR 1.96, 1.06–3.64, *p* = 0.032) and biphasic histology (HR 2.99, 1.54–5.80, *p* = 0.001) while recurrence was associated with biphasic histology (HR 2.17, 1.14–4.13, *p* = 0.019) and nodal involvement (HR 3.39, 1.75–6.57, *p* < 0.001). In the multivariable analysis, post-operative sarcopenia remained a significant predictor of death at follow-up even after the adjustment for the extension of the disease (HR 1.75, 1.02–3.03, *p* = 0.04) according to the pathological stage (pTNM, 8th edition).

### 3.4. Analysis of Pre-Operative Pulmonary Function

In addition, since sarcopenia may be strictly associated with pulmonary function, we analyzed the pre-operative pulmonary function for each patient in terms of forced expiratory volume in the first second (FEV1, % of predicted), forced vital capacity (FVC, % of predicted), total lung capacity (TLC, % of predicted), diffusion capacity of carbon monoxide (DLCO, % of predicted), and the maximal oxygen consumption (VO2max, mL/Kg/min). We found that the mortality at follow-up was significantly and non-linearly associated with FVC% (*p* = 0.002) (Figure 5a), with TLC% (*p* < 0.001) (Figure 5b), and with VO2max (*p* = 0.009) (Figure 5c).

In particular, FVC values between 60% and 87% were found to be protective (HR 0.2, IC 0.114–0.417) as well as TLC values between 60% and 90% (HR 0.1, IC 0.046–0.280) and VO2max values between 17 mL/Kg/min and 24 mL/Kg/min (HR 0.2, IC 0.072–0.565).

## 4. Discussion

Sarcopenia, defined as a progressive and generalized skeletal muscle disorder associated with an increased likelihood of adverse outcomes, including physical disability and mortality [1], has been recognized as a negative prognostic factor in several clinical settings. In particular, the low muscle mass is a marker of increased catabolic status and limited protein reserve, which is essential during stress periods such as major surgery and hospitalization [2]. For this reason, in recent times, the role of sarcopenia has been widely investigated in thoracic surgery: Although several studies have elucidated its prognostic influence on surgical outcomes in non-small cell lung cancer [10] and in lung transplantation [11,12,13], data on the role of sarcopenia in mesothelioma surgery are still lacking although the fact that patients affected by MPM have high rates of pre-sarcopenia and malnutrition is already established [18].

In our study, we aimed to assess the clinical association between pre- and post-operative sarcopenia and short-and long-term post-surgical outcomes in patients submitted to surgery for MPM. For the first time, we have investigated both the presence of pre- and post-operative sarcopenia, correlating both of them with an extensive analysis of clinical outcomes in addition to long-term survival and recurrence rates. Although a wide variety of tests and tools are now available for the assessment of sarcopenia both in clinical practice and research, we decided to use a simple, reliable, and non-invasive instrument as a CT scan without contrast as it can affect the muscle density [19]. Since abdominal CT scans are not always part of the common pre- and post-operative workup in thoracic patients, we decided to measure the densitometry of paravertebral muscles at the T12 level [20,21], assessing whether it can be used as a good surrogate in this population to overcome the problem of the unavailability of abdominal CT scans. In addition, not performing a volumetric evaluation of muscle composition, we wanted to assess an easy, feasible, and reproducible method in clinical practice without any particular software of analysis.

Talking about the outcomes, we found that patients with post-operative sarcopenia at the time of a CT scan have a significantly lower OS compared to non-sarcopenic ones (*p* = 0.03). A possible explanation of this finding is that surgery for MPM is always part, according to our center’s protocol, of a multimodality treatment consisting of neo-adjuvant CT and post-operative RT. Recently, it has been reported that patients with sarcopenia are at a significantly higher risk of developing platinum-related high-grade hematological toxicity [22] as well as increased RT toxicity [23,24]. The multimodal treatment could have played a cumulative effect, further debilitating patients with a persistent post-operative sarcopenic status resulting in lower overall survival. Furthermore, a lower response to systemic treatment with higher toxicity was already reported in sarcopenic patients with peritoneal carcinomatosis [25]; in our study, 100% of the patients underwent inductive CT as a part of our standard multimodality treatment, and a low chemotherapy response induced by sarcopenia could have negatively affected the post-surgical survival rate.

Secondly, it is known that skeletal muscle loss has a strong effect on the immune system, promoting an increase in pro-inflammatory cytokines, such as tumor growth factor (TGF-ß) and interleuchin-6 (IL-6), which play an important role in tumor growth, recurrence, and, consequently, in survival [26], even though, in our study a correlation between the cumulative incidence of relapse and sarcopenia was not found.

Again, the pro-inflammatory and hypercatabolic state induced by sarcopenia [26] has an impact not only on survival but also on post-operative complications, factors which may be closely related. In support of this, we demonstrated that the presence of pre-operative sarcopenia was significantly associated with a higher risk of developing post-operative complications (*p* = 0.04), according to the Clavien–Dindo Classification [27], in accordance with what has already been reported for NSCLC surgery [28].

In addition, we further investigated the patients’ performance status by analyzing their pre-operative respiratory function. The concept of “respiratory sarcopenia” has recently been defined as a whole-body sarcopenia and a low respiratory muscle mass followed by low respiratory muscle strength and/or deteriorated respiratory function [29]. Although there is still a lack of evidence in the specificity of the measures and their cut-off values, respiratory sarcopenia has already been assessed using respiratory functional parameters such as FVC, FEV1, maximal expiratory, and inspiratory pressure (MEP and MIP) [29,30]. In our analysis, we found that pre-operative values of FVC between 60 and 87%, TLC between 60 and 90%, and VO2max between 17 and 24 mL/min/Kg were protective from the risk of death.

All of these findings strongly suggest that the correct selection of the patient, trying to improve the performance and nutritional status, is mandatory before enrollment in multimodality treatment for malignant pleural mesothelioma.

To the best of our knowledge, only another available recent paper in current literature investigates this topic: according to our study, and also Verhoek et al. [31], who found sarcopenia to be a negative predictor of outcomes in surgically treated pleural mesothelioma but with some differences from our results. Unlike us, they found that patients affected by pre-operative sarcopenia had higher 3-year mortality while they did not evaluate the effect of post-operative sarcopenia. This difference in the results could have been due to a different technical methodology: they assessed the presence of sarcopenia by evaluating the muscle area at the T5 level, also using CT scans with contrast and without excluding patients submitted to surgery with palliative intent, which may have affected the results. Further studies on this topic are mandatory to standardize as much as possible the technique to assess the presence of sarcopenia in this population and to better understand molecular mechanisms that link sarcopenia to pleural mesothelioma. As already said, malnutrition and the consequent muscular loss, as well as cancers, induce a strong dysregulation of inflammasome components. The role of Interleuchin-6 (IL6), Tumor Necrosis Factor-alfa (TNF-alfa), C reactive protein (CRP), fibrinogen, and neutrophil/lymphocyte ratio (NLR) in MPM has been widely investigated by several authors [32,33,34,35,36,37,38] confirming that these molecules are not only overexpressed in this neoplasia, but they also have a negative prognostic impact. Inflammatory cells have important effects in the tumor microenvironment on tumor development and, as already performed for other forms of neoplasia [39], the investigation of the aforementioned molecular markers is mandatory also for pleural mesothelioma to provide significant information for prognostication [40,41]. As recently investigated for gastric cancers [42], we also acknowledge MPM for the need for future investigations, searching for molecular predictors of malnutrition and sarcopenia as they might represent targets for reversing a sarcopenic status before submitting a patient with pleural mesothelioma to surgery. Again, given the emergence of program cell death protein-1 (PD-1) pathway blockade as an effective therapeutic option in MPM, there is an increasing interest in which expression of program death ligand 1 (PD-L1) might be prognostic of clinical outcomes. To date, current evidence is still insufficient to draw any definitive conclusion on this topic [43], but future investigations are required to support the role of PD-L1 in MPM, correlating it to body composition parameters and performance status of the patients.

The limitations of this study are as follows: firstly, this is a monocentric retrospective study with relatively low numerosity and an inhomogeneous cohort of patients undergoing different therapeutic approaches. Secondly, thoracic muscles are constantly exercised by respiratory movements, and they may not atrophy, similar to abdominal muscles or quadriceps. The measurement of sarcopenia using thoracic muscles, although already reported by several authors [9,10] as a validated method, may be affected by some bias. Lastly, several studies have reported on the importance of the skeletal muscle index as an indicator of muscle loss, but it is usually computed at the level of the third lumbar vertebra. Thus, further studies on this group of patients, including abdominal scans, are expected to assess the clinical value also of this parameter [44].

## 5. Conclusions

In conclusion, our study shows for the first time that both pre- and post-operative sarcopenia have a negative influence on clinical outcomes in patients surgically treated for malignant pleural mesothelioma. This finding, through a non-invasive and accessible method such as a thorax CT scan without contrast, will be a useful tool in selecting the most suitable candidate for MPM surgery. Further studies using high numbers of patients, especially prospective and multi-centric, are required to investigate this topic and to standardize the definition of sarcopenia in these patients.

## Figures and Tables

**Figure 1 cancers-14-03699-f001:**
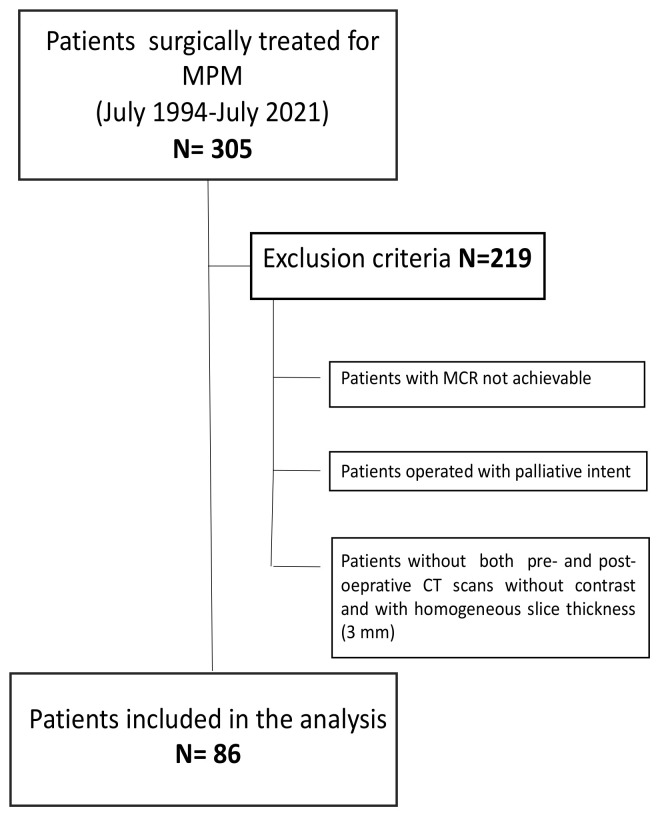
Inclusion and exclusion criteria. MPM: malignant pleural mesothelioma; MCR: macroscopic complete resection.

**Figure 2 cancers-14-03699-f002:**
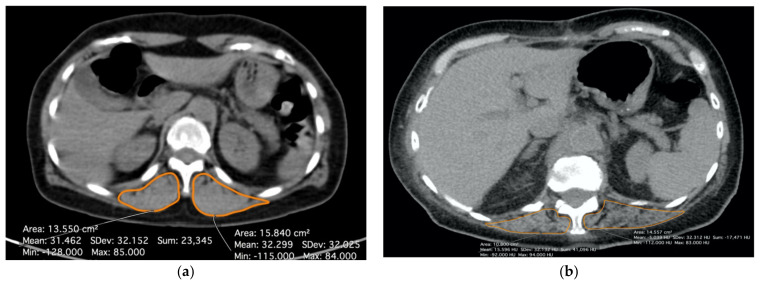
Radiological evaluation of sarcopenia at CT scan by measuring density of paravertebral muscles bilaterally (T12 level). (**a**) Patient without sarcopenia (muscular density > 30 HU); (**b**) Patient with sarcopenia (muscular density < 30 HU).

**Figure 3 cancers-14-03699-f003:**
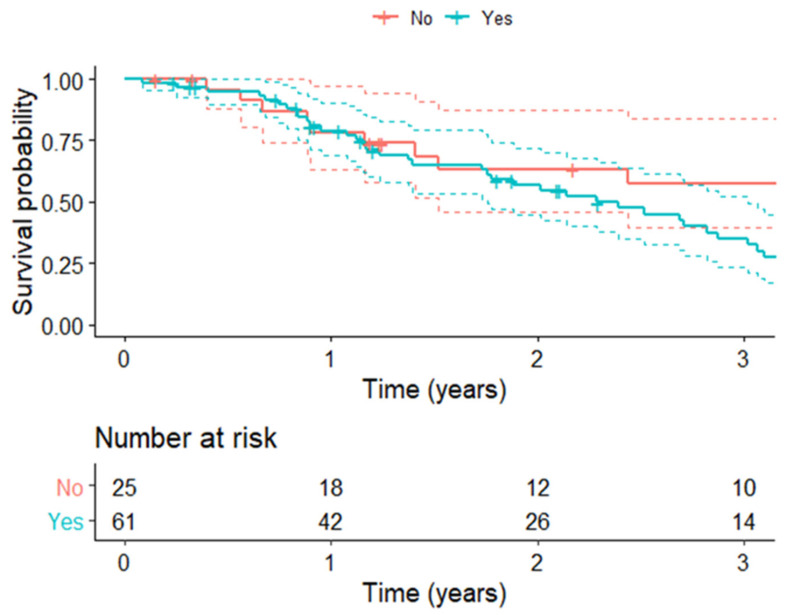
Overall survival rates of patients with post-operative sarcopenia (turquoise) compared to those without sarcopenia (red) (*p* = 0.03).

**Figure 4 cancers-14-03699-f004:**
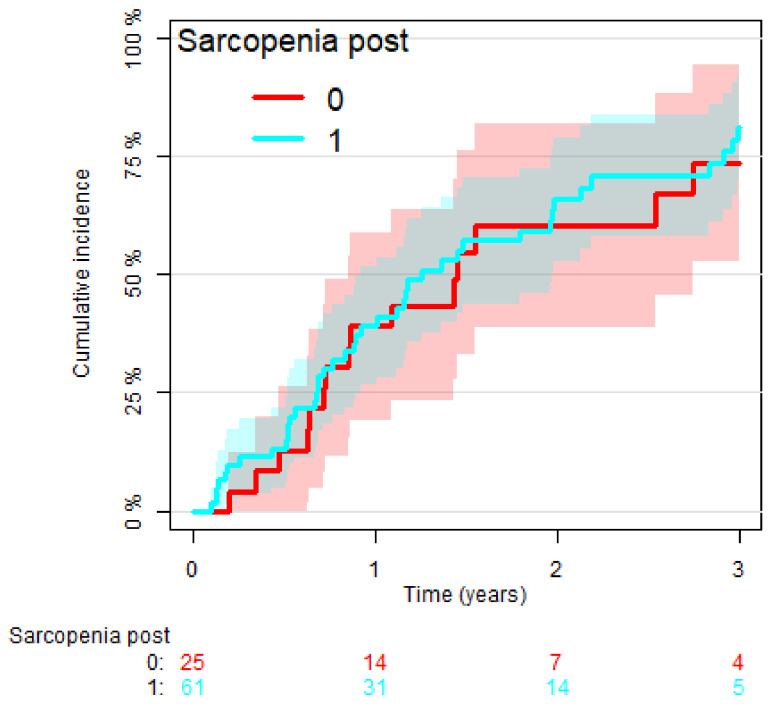
Cumulative incidence of relapse of patients with post-operative sarcopenia (turquoise) vs. no-sarcopenic (red) (*p* = 0.70).

**Figure 5 cancers-14-03699-f005:**
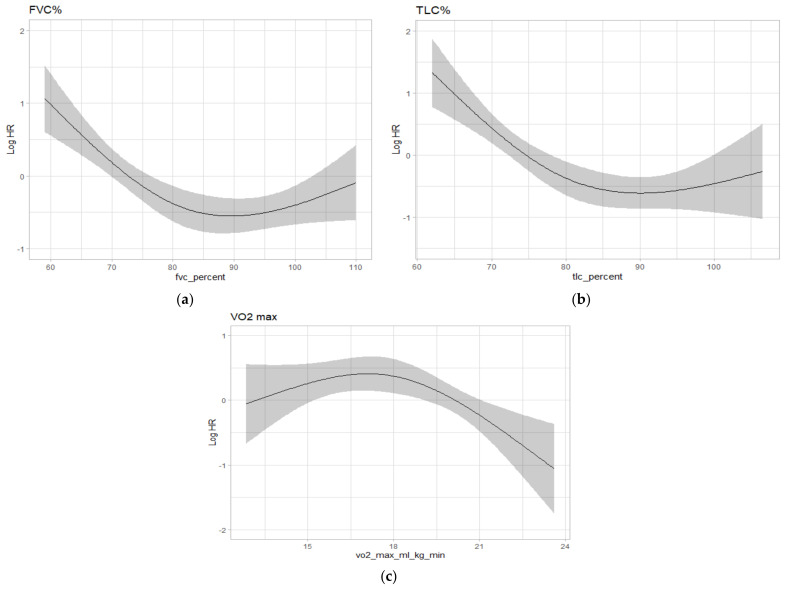
(**a**) relation between FVC (%) and the risk of mortality (Hazard Ratio reported in logarithmic scale); (**b**) relation between TLC (%) and the risk of mortality (Hazard Ratio reported in logarithmic scale); (**c**) relation between VO2max (mL/Kg/min) and the risk of mortality (Hazard Ratio reported in logarithmic scale).

**Table 1 cancers-14-03699-t001:** Demographic and clinical data of the overall population. Data are reported as median (I–III interquartile range) for continuous variables and as absolute numbers (relative frequencies) for categorical variables.

Variable	Overall (*n* = 86)
**Age** (years)	66 (62–71)
**Sex**	
F	21 (24%)
M	65 (76%)
**Smoker**	
No	34 (40%)
Yes	10 (12%)
Former	42 (48%)
**Asbestos exposure**	
No	34 (39.5%)
Yes	52 (60.5%)
**Surgery-side**	
Right	56 (65%)
Left	30 (35%)
**Type of surgery**	
PD/EPD	62 (72%)
EPP	24 (28%)
**Post-operative histology**	
Epithelioid	68 (79%)
Biphasic	16 (19%)
Desmoid	2 (2%)
**Neo-adjuvant CT**	
Yes	86 (100%)
No	0 (0%)
**Adjuvant RT**	
Yes	72 (84%)
No	14 (16%)
**pTNM (8th edition)**	
Ia	5 (6%)
Ib	43 (50%)
II	8 (9%)
IIIa	10 (12%)
IIIb	0 (0%)
IV	16 (19%)
Is	1 (1%)
CR	3 (3%)
**Pre-op FVC** (% predicted)	81 (70–92)
**Pre-op FEV1** (% predicted)	82 (74–92)
**Pre-op TLC** (% predicted)	79 (71–88)
**Pre-op DLCO** (% predicted)	66 (60–78)
**Pre-op VO2max** (mL/Kg/min)	17.9 (16.5–20.7)
**Pre-op BMI** (Kg/m^2^)	26.1 (24.5–28.4)
**Pre-operative sarcopenia**	
Yes (<30 HU)	57 (66%)
No (>30 HU)	29 (34%)
**Pre-op paravertebral muscle density** (HU)	27 (21–34)
**Post-operative sarcopenia**	
Yes (<30 HU)	63 (74%)
No (>30 HU)	23 (26%)
**Post-op paravertebral muscle density** (HU)	25 (17–30)
**Overall Survival** (months)	21 (11–38)
**Disease-free survival** (months)	13 (7–24)

F: female; M: male; CT: chemotherapy; RT: radiotherapy; PD: pleurectomy-decortication; EPP: extra-pleural pneumonectomy; EPD: extended pleurectomy-decortication; HU: Hounsfield Unit; FVC: forced vital capacity; TLC: total lung capacity; FEV1: forced expiratory volume in the 1st second; DLCO: diffusion capacity of carbon monoxide; VO2max: maximal oxygen consumption, BMI: body mass index; Is: in-situ; CR: complete remission; op: operative.

**Table 2 cancers-14-03699-t002:** Pre-operative sarcopenic vs. non-sarcopenic patients: demographic and clinical data. Data are reported as median (I–III interquartile range) for continuous variables and as absolute numbers (relative frequencies) for categorical variables.

Variable	Pre-Op Sarcopenia (*n* = 57)	No Sarcopenia (*n* = 29)	*p* Value
**Age** (years)	67 (63–71)	64 (57–67)	0.022
**Sex**			0.3
F	16 (28%)	5 (17%)	
M	41 (72%)	24 (83%)	
**Smoker**			0.5
No	22 (39%)	12 (41%)	
Yes	4 (7%)	4 (15%)	
Former	31 (54%)	13 (44%)	
**Asbestos exposure**			**0.037**
No	17 (30%)	17 (58%)	
Yes	40 (70%)	12 (42%)	
**Surgery side**			0.4
Right	39 (68%)	17 (59%)	
Left	18 (32%)	12 (41%)	
**Type of surgery**			0.14
PD/EPD	44 (77%)	18 (62%)	
EPP	13 (23%)	11 (38%)	
**Neo-adjuvant CT**			-
Yes	57 (100%)	29 (100%)	
No	0 (0%)	0 (0%)	
**Adjuvant RT**			0.5
Yes	47 (83%)	26 (90%)	
No	10 (17%)	3 (10%)	
**pTNM (8th edition)**			0.3
Ia	5 (9%)	0 (0%)	
Ib	28 (49%)	15 (52%)	
II	6 (10%)	2 (7%)	
IIIa	5 (9%)	5 (17%)	
IIIb	0 (0%)	0 (0%)	
IV	9 (16%)	7 (24%)	
Is	1 (2%)	0 (0%)	
CR	3 (5%)	0 (0%)	
**Pre-op FVC** (% predicted)	82 (70–91)	78 (70–94)	0.6
**Pre-op FEV1** (% predicted)	84 (73–92)	82 (75–94)	0.7
**Pre-op TLC** (% predicted)	80 (72–88)	78 (70–84)	0.4
**Pre-op DLCO** (% predicted)	66 (59–78)	68 (62–77)	0.6
**Pre-op VO2max** (mL/Kg/min)	17.4 (16.6–20.2)	19.5 (16.3–21.1)	0.2
**Pre-op BMI** (Kg/m^2^)	26.7 (24.7–28.7)	25.8 (24.5–26.7)	0.2
**Pre-op paravertebral muscle density** (HU)	24 (17–27)	38 (34–41)	**<0.001**

F: female; M: male; CT: chemotherapy; RT: radiotherapy; PD: pleurectomy-decortication; EPP: extra-pleural pneumonectomy; EPD: extended pleurectomy-decortication; HU: Hounsfield Unit; FVC: forced vital capacity; TLC: total lung capacity; FEV1: forced expiratory volume in the 1st second; DLCO: diffusion capacity of carbon monoxide; VO2max: maximal oxygen consumption, BMI: body mass index; Is: in-situ; RC: complete remission; op: operative. Bold *p* value (*p* < 0.05).

**Table 3 cancers-14-03699-t003:** Post-operative sarcopenic vs. non-sarcopenic patients: demographic and clinical data. Data are reported as median (I–III interquartile range) for continuous variables and as absolute numbers and relative frequencies for categorical variables.

Variable	Post-Op Sarcopenia (*n* = 61)	No Sarcopenia (*n* = 25)	*p* Value
**Age** (years)	68 (64–73)	64 (58–70)	**0.0014**
**Sex**			0.3
F	17 (28%)	5 (20%)	
M	44 (72%)	20 (80%)	
**Smoker**			0.059
No	27 (44%)	9 (35%)	
Yes	2 (3%)	5 (19%)	
Former	32 (53%)	11 (46%)	
**Asbestos exposure**			0.2
No	20 (32%)	17 (49%)	
Yes	41 (68%)	18 (51%)	
**Surgery side**			0.3
Right	43 (70%)	15 (59%)	
Left	18 (30%)	10 (41%)	
**Type of surgery**			0.6
PD/EPD	45 (74%)	17 (69%)	
EPP	16 (26%)	8 (31%)	
**Neo-adjuvant CT**			-
Yes	61 (100%)	25 (100%)	
No	0 (0%)	0 (0%)	
**Adjuvant RT**			0.3
Yes	50 (82%)	22 (89%)	
No	11 (18%)	3 (11%)	
**pTNM (8th edition)**			0.3
Ia	7 (11%)	0 (0%)	
Ib	30 (50%)	13 (51%)	
II	7 (11%)	2 (8%)	
IIIa	7 (11%)	3 (13%)	
IIIb	0 (0%)	0 (0%)	
IV	8 (13%)	6 (23%)	
Is	0 (0%)	1 (5%)	
CR	2 (4%)	0 (0%)	
**Pre-op FVC** (% predicted)	80 (70–87)	81 (70–98)	0.6
**Pre-op FEV1** (% predicted)	83 (73–90)	82 (75–97)	0.5
**Pre-op TLC** (% predicted)	80 (70–87)	79 (74–88)	>0.9
**Pre-op DLCO** (% predicted)	66 (54–77)	68 (61–82)	0.3
**Pre-op VO2 max** (mL/Kg/min)	17.3 (16.6–20.4)	18.8 (16.6–20.4)	0.2
**Pre-op BMI** (Kg/m^2^)	26.2 (24.3–28.4)	25.9 (24.7–28.4)	0.2
**Pre-op paravertebral muscle density** (HU)	21 (16–27)	34 (30–41)	**<0.001**

F: female; M: male; CT: chemotherapy; RT: radiotherapy; PD: pleurectomy-decortication; EPP: extra-pleural pneumonectomy; EPD: extended pleurectomy-decortication; HU: Hounsfield Unit; FVC: forced vital capacity; TLC: total lung capacity; FEV1: forced expiratory volume in the 1st second; DLCO: diffusion capacity of carbon monoxide; VO2max: maximal oxygen consumption, BMI: body mass index; Is: in-situ; CR: complete remission; op: operative. Bold *p* value (*p* < 0.05).

**Table 4 cancers-14-03699-t004:** Post-operative outcomes (pre-operative sarcopenic vs. non-sarcopenic patients). Data are reported as median (I–III interquartile range) for continuous variables and as absolute numbers and relative frequencies for categorical variables.

Variable	Pre-Op Sarcopenia (*n* = 57)	No Sarcopenia (*n* = 29)	*p* Value
**Length of hospital stay** (d)	14 (11–21)	12 (10–20)	0.9
**Post-operative complications**			**0.04**
Yes	37 (65%)	12 (41%)	
No	20 (35%)	17 (59%)	
**Clavien–Dindo classification**			0.14
1 + 2	42 (73%)	15 (51%)	
3 + 4 + 5	15 (27%)	14 (49%)	

d: days. Bold *p* value (*p* < 0.05).

## Data Availability

The data presented in the study are available on request from the corresponding author. The data are not publicly available due to ethical reasons.

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
