# Peer review of "Sarcopenia as a Predictor of Short- and Long-Term Outcomes in Patients Surgically Treated for Malignant Pleural Mesothelioma"

_cancers, 2022, doi:10.3390/cancers14153699_

Round 1

Reviewer 1 Report

In the present manuscript “Sarcopenia as A Predictor of Short- and Long-term Outcomes in Patients Surgically Treated for Malignant Pleural Mesothelioma”-Faccioli et al investigated the indeed important and still unclear issue of sarcopenia in malignant pleural mesothelioma (MPM). This is an interesting topic. However, some major points have to be addressed before publication of the present manuscript can be recommended from my side

CONSORT diagram considering study in- and exclusion is missing

Please comment on the cut off value (30HU). Why did you choose this value?

Please clarify, what kind of sarcopenia was prognostic also in the abstract. Was it the pre- or the postoperative one?

When talking about sarcopenia, you should also discuss the role of IL6 in MPM, one of the major players when it comes to disease progression, sarcopenia, pro-inflammatory status and weight loss and poor performance status. Please also discuss this important issue. (compare British Journal of Cancer 1998 Nakano et al)

Sarcopenia should be correlated to the performance status of the patient

Sarcopenia must be correlated to the inflammatory status of the patient

Also compare comment regarding IL6.

CRP, fibrinogen and the NLR are well-characterized and important prognostic markers. Also compare Annals of Surgery 2012 Ghanim et al, regarding CRP as predictive marker for surgery in MPM, and similar to these results, British Journal of Cancer 2014 Ghanim et al. considering fibrinogen. Furthermore, also compare European Journal of Cardio-Thoracic Surgery 2022 by Greb et al and Takamori et al Ann Surg Oncol 2018 with regard to the CAR. In addition, also get back to the Meta-Analysis of Chen et al in Oncotarget 2017 about the NLR or Pinato et al J Thorac Oncol. 2012 analyzing GPS and NLR

Please also compare the review articles “Inflammation in malignant mesothelioma—friend or foe” by Linton et al and more recently “Biomarkers for Malignant Pleural Mesothelioma—A Novel View on Inflammation” by Vogl et al

From my point of view, it is mandatory to analyze the inflammatory status of a MPM patient when talking about sarcopenia. Also compare e.g. recent publication of U T Hacker et al (Ann Oncol. 2022)

Can you compare (at least some) inflammatory parameters to the presence of sarcopenia, e.g. NLR, CRP, fibrinogen, Albumin or the Glasgow Prognostic score.

What about PDL1 and immune therapy? This is an interesting point with rising clinical relevance for MPM patients. Was there a correlation between PDL1 expression and sarcopenia?

Sarcopenia should be correlated to the disease stage and tumor burden/tumor volume

Was sarcopenia maybe only a reflection of advanced disease? This information is important and should at least be discussed. Multivariate analyses would be better to exclude that sarcopenia and its prognostic value are only indirect surrogates of bad outcome

Author Response

In the present manuscript “Sarcopenia as A Predictor of Short- and Long-term Outcomes in Patients Surgically Treated for Malignant Pleural Mesothelioma”-Faccioli et al investigated the indeed important and still unclear issue of sarcopenia in malignant pleural mesothelioma (MPM). This is an interesting topic. However, some major points have to be addressed before publication of the present manuscript can be recommended from my side

CONSORT diagram considering study in- and exclusion is missing

We thank the reviewer for this useful suggestion. A diagram with the inclusion and exclusion criteria is provided as Figure 1.

Please comment on the cut off value (30HU). Why did you choose this value?

We thank the Reviewer for giving us the chance to better highlight this aspect. The cut-off has been selected according to the literature. In fact, several studies demonstrated that values of muscle density <30 Hu reflect low attenuation muscles. To further support this choice, we have added four references (Goodpaster BH, Thaete FL, Kelley DE. Composition of skeletal muscle evaluated with computed tomography. Ann N Y Acad Sci. 2000; 904:18–24. https://doi.org/10.1111/j.1749-6632.2000.tb06416.x PMID: 10865705; Aubrey J, Esfandiari N, Baracos VE, Buteau FA, Frenette J, Putman CT, et al. Measurement of skeletal muscle radiation attenuation and basis of its biological variation. Acta Physiol (Oxf). 2014; 210:489– 497. https://doi.org/10.1111/apha.12224 PMID: 24393306; Lee S, Kuk JL, Davidson LE, Hudson R, Kilpatrick K, Graham TE, et al. Exercise without weight loss is an effective strategy for obesity reduction in obese individuals with and without type II diabetes. J Appl Physiol (1985). 2005; 99:1220–1225.; Kim et al. Prognostic significance of radiodensity-based skeletal muscle quantification using preoperative CT in resected non-small cell lung cancer doi 10.21037/jtd-20-2344) and modified one sentence in the materials and methods, which now reads “According to the literature, a value of 30 HU was assessed as a cut-off to define a patient as sarcopenic [14-17].”

Please clarify, what kind of sarcopenia was prognostic also in the abstract. Was it the pre- or the postoperative one?

We thank the reviewer to better point out this important aspect. We have specified also in the simple summary (line 25) which kind of sarcopenia respectively affects the ouctomes considered in our study. In the abstract too (lines 38-41) we have pointed out that in our analysis post-operative sarcopenia had an influence on overall survival while pre-operative sarcopenia was correlated with a higher frequency of post-operative complications.

When talking about sarcopenia, you should also discuss the role of IL6 in MPM, one of the major players when it comes to disease progression, sarcopenia, pro-inflammatory status and weight loss and poor performance status. Please also discuss this important issue. (compare British Journal of Cancer 1998 Nakano et al)

We appreciate this suggestion. We have added considerations regarding the role of IL-6 in MPM referring to the suggested reference. Please see in the discussion session.

Sarcopenia should be correlated to the performance status of the patient

This is a very important point. We agree with the reviewer: sarcopenia should be correlated with the performance status of our patients. Unfortunately, we do not have objective data (like ECOG or Karnofsky scale) for all the subjects enrolled in this study to perform a formal analysis in which sarcopenia and performance status could be correlated. In a future study we will certainly provide also this information.

Sarcopenia must be correlated to the inflammatory status of the patient

      We strongly appreciate this suggestion. Unfortunately, we do not have adequate pre- and post-operative data on the inflammatory status for all the patients enrolled in this study, as the aim of the present investigation was mainly focused on clinical aspects. Despite this, we completely agree with the reviewer on the importance of this aspect and it will be the aim of our future investigations on this topic. As we pointed out in our discussion “We acknowledge the need for a future investigation, searching for molecular predictors of malnutrition and sarcopenia as they might represent targets for reversing a sarcopenic status before submitting a patient with pleural mesothelioma to surgery”

Also compare comment regarding IL6. 

Thank you for this suggestion. We have added comment regarding IL6. Please see the discussion session.

CRP, fibrinogen and the NLR are well-characterized and important prognostic markers. Also compare Annals of Surgery 2012 Ghanim et al, regarding CRP as predictive marker for surgery in MPM, and similar to these results, British Journal of Cancer 2014 Ghanim et al. considering fibrinogen. Furthermore, also compare European Journal of Cardio-Thoracic Surgery 2022 by Greb et al and Takamori et al Ann Surg Oncol 2018 with regard to the CAR. In addition, also get back to the Meta-Analysis of Chen et al in Oncotarget 2017 about the NLR or Pinato et al J Thorac Oncol. 2012 analyzing GPS and NLR

We appreciate this suggestion. We have added considerations regarding inflammation based prognostic indeces in MPM adding the suggested references. Please see the discussion session

Please also compare the review articles “Inflammation in malignant mesothelioma—friend or foe” by Linton et al and more recently “Biomarkers for Malignant Pleural Mesothelioma—A Novel View on Inflammation” by Vogl et al

We appreciate this suggestion. We have added considerations regarding inflammation based prognostic indeces in MPM adding the suggested references. Please see the discussion session

From my point of view, it is mandatory to analyze the inflammatory status of a MPM patient when talking about sarcopenia. Also compare e.g. recent publication of U T Hacker et al (Ann Oncol. 2022)

We appreciate this suggestion. We have added considerations regarding inflammation based prognostic indeces in MPM adding the suggested references. Please see the discussion session.

Can you compare (at least some) inflammatory parameters to the presence of sarcopenia, e.g. NLR, CRP, fibrinogen, Albumin or the Glasgow Prognostic score.

We totally agree with the reviewer: this is a very important point. Unfortunately, as already specified in a previous reply, we do not have data on inflammatory parameters in our population as this study is focused on clinical parameters but this will certainly be the aim of our next studies on this topic.

What about PDL1 and immune therapy? This is an interesting point with rising clinical relevance for MPM patients. Was there a correlation between PDL1 expression and sarcopenia?

We really appreciate the opportunity to better point out this aspect with this question. The first point is that at our Institution, we search the PD-L1 expression status for every patient submitted to surgery for MPM only in recent years; for this reason the PD-L1 expression is not available for all the 86 patients enrolled in the analysis and, because of this, we are not able to provide a formal statystical analysis which correlates the PD-L1 expression to the sarcopenia. In addition, we were not able to find studies in current literarature wich point out a possible correlation between PD-L1 and body composition in MPM. Certainly, as specified in the discussion, this is very interesting suggestion for future investigations since the increasing interest on PD-L1 as a prognostic factor in MPM.

Sarcopenia should be correlated to the disease stage and tumor burden/tumor volume

We thank the reviewer for this comment as it is an important aspect to expand. We performed, as suggested also in the next question, a multivariate analysis to correlate sarcopenia to the stage of the disease and we found that sarcopenia remained a predictor of death even after this adjustement.

Was sarcopenia maybe only a reflection of advanced disease? This information is important and should at least be discussed. Multivariate analyses would be better to exclude that sarcopenia and its prognostic value are only indirect surrogates of bad outcome 

We really appreciate the opportunity to better point out this aspect with this question. Sarcopenia can be considered a reflection of advanced disease: we know that neoplastic patients, especially at an advanced stage of their disease, have a worse performance status associated with cachexia which in many cases is strictly correlated with a sarcopenic status. In our study we evaluated the presence of pre- and post-operative sarcopenia simply measuring the area of paravertebral muscles at the CT scans. Other tools (such as hand grip test or chair test), which could have provided a more realistic picture of muscular strenght, were not measured in our population but it is a proper suggestion for next investigation.

However, as suggested by the reviewer, we performed a multivariate analyses and sarcopenia remained a significant predictor of death at follow-up even after the adjustment for the pathological stage (pTNM 8Th edition). This evidence supports the fact that sarcopenia is a real prognostic factor and not only a surrogate of negative outcomes. We have added this finding also in the results.

Reviewer 2 Report

1) in the methods section. (2.2) the authors declare that a circular ROI was used but figure 1 shows an hand-made ROI segmentation 

2) sarcopenia is here defined according to HU values (<30) but the skeletal muscle index (SMI) is considered more appropriate 

3)you defined as sarcopenia what really is a fat infiltration of muscle , which is not sarcopenia; please consider using the term muscle adiposity or fatty muscle

4) is sarcopenia a causative factor for negative outcomes or simply a factor associated to negative outcomes?

Author Response

1) in the methods section. (2.2) the authors declare that a circular ROI was used but figure 1 shows an hand-made ROI segmentation 

We would like to thank the reviewer for noticing the oversight in the manuscript and apologize for it. All values were extracted by free-hand ROIs of the paraspinal muscles at the level of the 12th thoracic vertebra as demonstrated in figure 1. We have corrected the corresponding sentence in the manuscript, which now reads “For each patient the mean Hounsfield Unit (HU) value of the bilateral paravertebral muscles at the level of 12th thoracic vertebra, using a free-hand region of interest, was calculated.”

2) sarcopenia is here defined according to HU values (<30) but the skeletal muscle index (SMI) is considered more appropriate 

We agree with the Reviewer that the skeletal muscle index (SMI) represents another very important parameter to assess muscle loss at imaging. Nevertheless, we consider justified our choice to evaluate only muscle density at CT in this study. In fact, the SMI is usually computed at the level of the third lumbar vertebra (Mc Govern et al. https://doi.org/10.1002/jcsm.12831), which was not available in our population. Although, a few recent studies tested other vertebral levels [Matsuyama et al https://doi.org/10.1016/j.nut.2021.111475], the cut-off values usually applied (41 cm2/m2) refers to the third lumbar vertebra and different measurements would have required a further validation going beyond the scope of this project.

Considering also that the guidelines of the international task force for sarcopenia mention the application of imaging (i.e., DXA, CT, and MR) but without clearly specifying the role of each of these techniques and/or radiological parameters such as muscle density, SMI, or chemical-shifted encoded proton density fat fraction by MR, to date there is no specific indication about which technique/parameter to use as gold standard. Certainly, further studies on this group of patients are expected to evaluate also the SMI, providing new insights. In the limitations, we have added the reference of Mc Govern et al and the following sentence: “Last, several studies reported the importance of the skeletal muscle index as indicator of muscle loss. Thus, further studies on this type of patients are expected to assess the clinical value of this parameter.

3) you defined as sarcopenia what really is a fat infiltration of muscle, which is not sarcopenia; please consider using the term muscle adiposity or fatty muscle

We thank the Reviewer for this valuable comment. We agree that terminology is very important. Nevertheless, considering that sarcopenia is defined as “an age-associated loss of skeletal muscle function and muscle mass” [Dent et al. International clinical practice guidelines for sarcopenia (ICFSR): screening, diagnosis and management. J Nutr Health Aging. 2018;22(10):1148-1161], we believe that the applied terms can be considered appropriate. In fact, the fat infiltration occurring in aging muscles/muscles altered by diseases represents part of the muscle loss since muscle fibers are replaced by atrophic/fatty areas. As mentioned in reply#2 further studies including additional parameters such as the SMI may provide a more comprehensive overview of the sarcopenia/muscle loss occurring in these patients.

4) is sarcopenia a causative factor for negative outcomes or simply a factor associated to negative outcomes?

We really appreciate this comment as it is a valid aspect to better point out. To better clarify this, we performed a multivariable analysis and sarcopenia remained a significant predictor of death at follow-up even after the adjustment for the pathological stage (pTNM 8Th edition). This evidence supports the fact that sarcopenia is a real prognostic factor and not only a surrogate of negative outcomes. We have specified this point also in the manuscript.

Reviewer 3 Report

This is an interesting report on the clinical significance of sarcopenia in patients with malignant pleural mesothelioma. 

Of the 305 patients with malignant pleural mesothelioma during the period, only 86 patients were included in the analysis. The patient selection should be summarized in the flow diagram which clearly describes the patient selection.

The details of the treatment of the patients are described in detail, but there is no description of rehabilitation intervention. The general policy of rehabilitation intervention for the patients should be added.

Overall, I would think this is a very valuable report and contains useful information.

Author Response

This is an interesting report on the clinical significance of sarcopenia in patients with malignant pleural mesothelioma. 

Of the 305 patients with malignant pleural mesothelioma during the period, only 86 patients were included in the analysis. The patient selection should be summarized in the flow diagram which clearly describes the patient selection.

We really thank the reviewer for this suggestion. We have provided a diagram which described the patients’ selection in the method sessione (Figure 1).

The details of the treatment of the patients are described in detail, but there is no description of rehabilitation intervention. The general policy of rehabilitation intervention for the patients should be added.

We appreciate the opportunity to point out this important aspect. According to our policy, all the patients submitted to surgery for malignant pleural mesothelioma at our Instituition are evaluated by a physiatrist the day after the intervention who sets a specific respiratory rehabilitation program tailored on every patient’s conditions. After this, the patients are followed the subsequent days by a physiotherapist who educate the patients on these specific exercises to improve physical and respiratory ability especially with use of incentive spirometer. The patients are also encouraged to continue these exercises after the discharge. We have added these details also in the manuscript in the method session.

Overall, I would think this is a very valuable report and contains useful information.

We really appreciate this final comment from the reviewer!